# Creating two-dimensional solid helium via diamond lattice confinement

Weitong Lin [1,12], Yiran Li [2,12], Sytze de Graaf [3], Gang Wang[4], Junhao Lin [4], Hui Zhang[5], Shijun Zhao [1], Da Chen[6], Shaofei Liu[1], Jun Fan [7], Bart J. Kooi [3], Yang Lu [1,8], Tao Yang [7] ✉, Chin-Hua Yang[9,10], Chain Tsuan Liu [7] & Ji-jung Kai [1,11] ✉

The universe abounds with solid helium in polymorphic forms. Therefore, exploring the allotropes of helium remains vital to our understanding of nature. However, it is challenging to produce, observe and utilize solid helium on the earth because high-pressure techniques are required to solidify helium. Here we report the discovery of room-temperature two-dimensional solid helium through the diamond lattice confinement effect. Controllable ion implantation enables the self-assembly of monolayer helium atoms between {100} diamond lattice planes. Using state-of-the-art integrated differential phase contrast microscopy, we decipher the buckled tetragonal arrangement of solid helium monolayers with an anisotropic nature compressed by the robust diamond lattice. These distinctive helium monolayers, in turn, produce substantial compressive strains to the surrounded diamond lattice, resulting in a large-scale bandgap narrowing up to ~2.2 electron volts. This approach opens up new avenues for steerable manipulation of solid helium for achieving intrinsic strain doping with profound applications.

Helium is the second most abundant chemical element in the universe. For example, hexagonal close-packed metallic helium is expected to dominate the secular cooling process of white dwarf stars[1]. At low temperatures, helium can become archetypal quantum crystals with exotic physical properties, such as second sound[2], superfluidity[3], and giant plasticity[4]. Stabilizing solid helium in two dimensions is of fundamental interest for exploring quantum atomic physics and exotic states of matter[5–7]. Therefore, exploring the allotropes of helium remains vital to our understanding of nature. However, our knowledge about helium, especially in the solid state, is still quite limited because high-pressure techniques are necessarily required to solidify

helium[8–10]. Owing to its quantum crystalline nature, helium only solidifies under high pressures, such as more than around 11.5 GPa at room temperature[10]. Over the past decades, synchrotron X-ray techniques coupled with diamond anvil cells have been used to probe the structure of bulk solid helium[11,12] and helium compounds[13,14]. Furthermore, two-dimensional solid helium adsorbed on the surface of graphite has been extensively studied below 7.4 K as a simple model for understanding phase transitions[15–19]. However, direct observation and steerable manipulation of two-dimensional solid helium at ambient temperature have hitherto been elusive because of the lack of practicable high-pressure devices.

[1]Department of Mechanical Engineering, City University of Hong Kong, Hong Kong, China. [2]School of Materials Science and Engineering, Shanghai University, Shanghai, China. [3]Zernike Institute for Advanced Materials, University of Groningen, 9747 AG Groningen, The Netherlands. [4]Department of Physics, Southern University of Science and Technology, Shenzhen, China. [5]Energy Geoscience Division, Lawrence Berkeley National Laboratory, Berkeley, CA, USA. [6]School of Energy and Environment, Southeast University, Nanjing, China. [7]Department of Materials Science and Engineering, City University of Hong Kong, Hong Kong, China. [8]Nano-Manufacturing Laboratory (NML), Shenzhen Research Institute of City University of Hong Kong, Shenzhen, China. [9]Department of Biomedical Engineering and Environmental Science, National Tsing Hua University, Hsinchu, Taiwan. [10]Department of Radiology, Taoyuan General Hospital, Taoyuan, Taiwan. [11]Centre for Advanced Nuclear Safety and Sustainable Development, City University of Hong Kong, Hong Kong, China. [12]These authors contributed equally: Weitong Lin, Yiran Li. ✉e-mail: taoyang6-c@my.cityu.edu.hk; jijkai@cityu.edu.hk

Alternatively, the insolubility of helium in solids provides a feasible route to fossilize two-dimensional solid helium specimens through the lattice confinement effect. The concept of helium trapped in a "condensed state" at room temperature has been previously investigated for helium-implanted metals mainly in relation to damage in materials for nuclear reactors[20]. Helium trapped in silicon has been also widely investigated and the introduction of helium nanobubbles has found applications in silicon technology[21–24]. Moreover, in the case of negligible availability of vacancies, implanted helium ions can spontaneously precipitate into nanoscale helium platelets[25–27], where helium atoms are confined in a two-dimensional structure. In theory, high static pressures generated by the surrounding crystal lattice potentially allow us to stabilize two-dimensional solid helium at room temperatures[28], the mysterious structure of which is expected to be experimentally identified by the atomic-resolution scanning transmission electron microscopy (STEM). To this end, diamond, as the hardest material in nature[29], has been considered as one of the most promising containers for achieving two-dimensional solid helium with sufficient pressures. More intriguingly, we believe that two-dimensional solid helium could give rise to unique strain engineering responses of diamond lattice, contributing to the development of diamond devices with tuneable bandgap characters and potentially unusual functional properties.

Although theoretically feasible, it fails in practice to controllably achieve two-dimensional solid helium until now. A long-standing challenge is to suppress the radiation-induced amorphization[30] and graphitization[31] of the diamond lattice during the helium ion implantation. Such structural instability induced by irradiation damage can substantially reduce the stiffness of diamond[32], resulting in the preferential formation of helium nanobubbles or significantly lower pressure inside platelets. On the other hand, helium diffusion is significantly limited in the diamond lattice due to the short and strong $sp^3$-bonded carbon structure[33], which seriously inhibits the formation of helium platelets. In the present work, we overcome these dilemmas by increasing the helium ion implantation temperature to 1573 K, simultaneously enhancing the point defect recombination rate and helium diffusion rate in the diamond lattice.

## Results

### S/TEM observations

Figure 1a depicts a schematic of the apparatus that we elaborately designed to perform ion implantation into diamonds at such an ultra-high temperature. As such, single-crystalline <100> diamonds were implanted with 275 keV $^4$He$^+$ ions at 1573 K to a fluence of $6.4 \times 10^{16}$ ions cm$^{-2}$ in a high-vacuum chamber (see Methods and Supplementary Fig. 1a). The distribution of helium atoms inside diamond was predicted by using stopping and range of ions in matter code (see Supplementary Fig. 1b). After that, we prepared high-quality cross-sectional diamond specimens with a focused ion beam (FIB) technique (see Supplementary Fig. 2) for further transmission electron microscopy (TEM) observation.

The Fresnel contrast TEM characterization, as displayed in Fig. 1b, shows a high density of helium precipitates at depths ranging from 600 to 700 nm underneath the surface of the diamond specimen

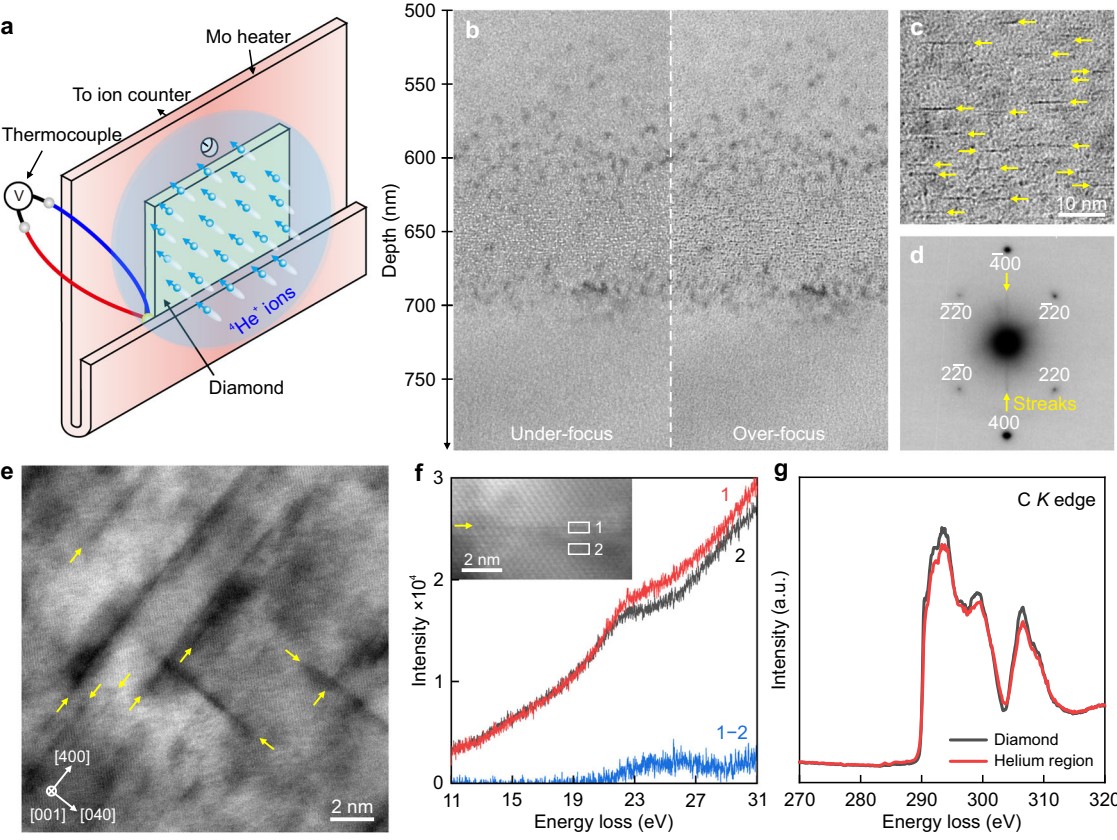

**Fig. 1 | High-temperature $^4$He$^+$ ion implantation. a** Schematic of $^4$He$^+$ ion implantation into diamond at high temperature. **b** Fresnel contrast TEM images show helium ion implantation region in the under-focus ($\Delta f = -400$ nm) and over-focus ($\Delta f = +400$ nm) conditions. **c** Higher-magnification Fresnel contrast TEM image demonstrates helium platelets, which are highlighted by yellow arrows, in the over-focus ($\Delta f = +200$ nm) condition. A two-beam condition $\mathbf{g} = 400$ was used. **d** Selected area electron diffraction pattern of the helium region confirms the formation of {100} helium platelets. **e** HAADF STEM image illustrates {100} helium platelets. **f** Low-loss electron energy-loss spectra obtained on (area 1) and off (area 2) the platelet. **g** Core-loss electron energy-loss spectra show carbon $K$ edge from the helium ion implantation region and the pristine diamond. Source data are provided as a Source Data file.

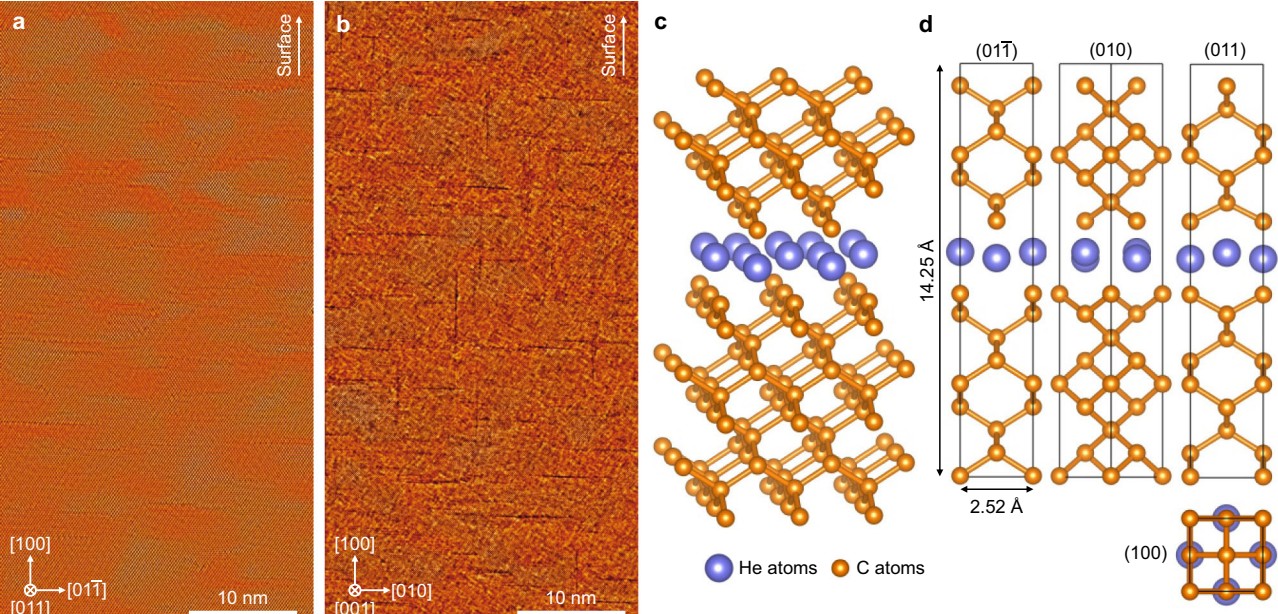

**Fig. 2 | Helium platelets in diamond.** Large-field-of-view iDPC STEM images under the **a** [011] and **b** [001] zone axes illustrate helium platelets in diamond. **c** Atomic structural model of {100} two-dimensional solid helium confined by the diamond lattice. **d** Atomic positions of two-dimensional solid helium in the unit cell, which has the dimensions of $a = 2.52$ Å, $b = 2.52$ Å and $c = 14.25$ Å.

(hereafter referred to as the helium region). The distribution of helium precipitates is well consistent with the theoretical prediction (see Supplementary Fig. 1b). A higher-magnification Fresnel contrast TEM image (see Fig. 1c) highlights the formation of helium platelets (indicated by yellow arrows) within the helium region by the self-assembly of helium atoms into a two-dimensional layer. As expected, the corresponding selected area electron diffraction pattern (see Fig. 1d) exhibits the streaks from the thin disk-shaped {100} helium platelets. Additionally, diffraction spots from the crystalline diamond can be observed, whereas none of the diffraction spots from the amorphous diamond or graphite are detected. It suggests that by rendering helium interstitial atoms mobile but retaining the diamond structure without amorphization and graphitization, helium atoms are self-assembled into two dimensions between the {100} planes of the diamond lattice. We further explore whether the diamond orientation determines the crystallographic orientation of helium platelets. A polycrystalline diamond produced by chemical vapor deposition was implanted with $^4$He$^+$ ions at 1573 K. TEM characterizations confirmed the presence of {100} helium platelets in a [3$\bar{1}$1]-oriented diamond grain (see Supplementary Fig. 3). Such evidence favored the {100} habit plane of these helium platelets inside the diamond lattice.

The presence of {100} helium platelets was also supported by high-angle annular dark-field (HAADF) STEM imaging. Figure 1e shows helium platelets with a darker contrast in the diamond specimen because of the relatively low atomic number of helium. Furthermore, we performed spatially resolved electron-energy loss spectroscopy (EELS) measurements on a helium platelet and compared it with the spectrum taken from the adjacent matrix (see Fig. 1f). An edge is visible at ~25.0 eV from the helium platelet that can be attributed to the combination of helium $K$-edge and platelet-enhanced surface plasmon of the crystalline diamond. It further confirms that the helium platelet was confined by the surrounding diamond lattice. Moreover, we compared the core-loss spectra from the helium region and the pristine diamond (see Methods). As shown in Fig. 1g, intense peaks at around 294, 299, and 306 eV were observed that could be readily assigned as transitions from 1s to $\sigma^*$ states of $sp^3$-bonded diamond structures. By contrast, we did not observe the peak at 285.5 eV, which corresponds to the transition from the 1s to $\pi^*$ states of graphite or

amorphous carbon, supporting that the amorphization and graphitization of diamond lattice were successfully suppressed during our well-designed high-temperature ion implantation process.

## Determining helium atoms arrangements inside platelets

We next wish to decipher the atomic arrangements of helium platelets at room temperature by using integrated differential phase contrast (iDPC) STEM imaging. The schematic ray diagram for iDPC STEM is shown in Supplementary Fig. 4a. The aberration-corrected electron microscope employs an annular detector segmented into four quadrants (DPC1-4) to collect coherently scattered electrons. Such a unique technique has good capability for imaging various light elements[34], even including the lightest hydrogen[35], with atomic resolution and a good signal-to-noise ratio. As shown in Supplementary Fig. 4b, the microstructure of the pristine type Ib diamond was carefully examined by iDPC STEM imaging. We ruled out the presence of voidites[29] or nitrogen platelets[36] that might influence the observation of helium platelets. In Fig. 2a, b, wide-field iDPC STEM images viewed from the [011] and [001] zone axes demonstrate that helium platelets embedded in the diamond lattice exhibit an atomically thin and two-dimensional structure. Helium platelets have a mean diameter of 4.6 nm (see diameter distribution in Supplementary Fig. 5a) and an average opening of ~0.24 nm. According to the statistical results from iDPC STEM imaging (see Supplementary Fig. 5b), the relative fraction of platelets on the (100) and (010) habit planes is about 2:1. It suggests that helium platelets in diamond have a strongly anisotropic distribution. Additionally, helium platelets were characterized through depth-sectioning iDPC STEM imaging that could reveal three-dimensional information in the diamond specimen. By changing probe defocus values, helium platelets at different heights within the diamond TEM specimen can be observed (see Supplementary Fig. 6). To determine the helium pressure inside platelets, we built up the diamond lattice with the inserted two-dimensional solid helium over a platelet spacing range from 1.07 to 2.14 nm (see Supplementary Fig. 7a). After full relaxation of atomic positions via density functional theory (DFT) calculations (see Methods), we obtained the supercell stress in different directions (see Supplementary Fig. 7b). It demonstrates that the interlayered pressure, i.e., $P_{zz}$, increases with

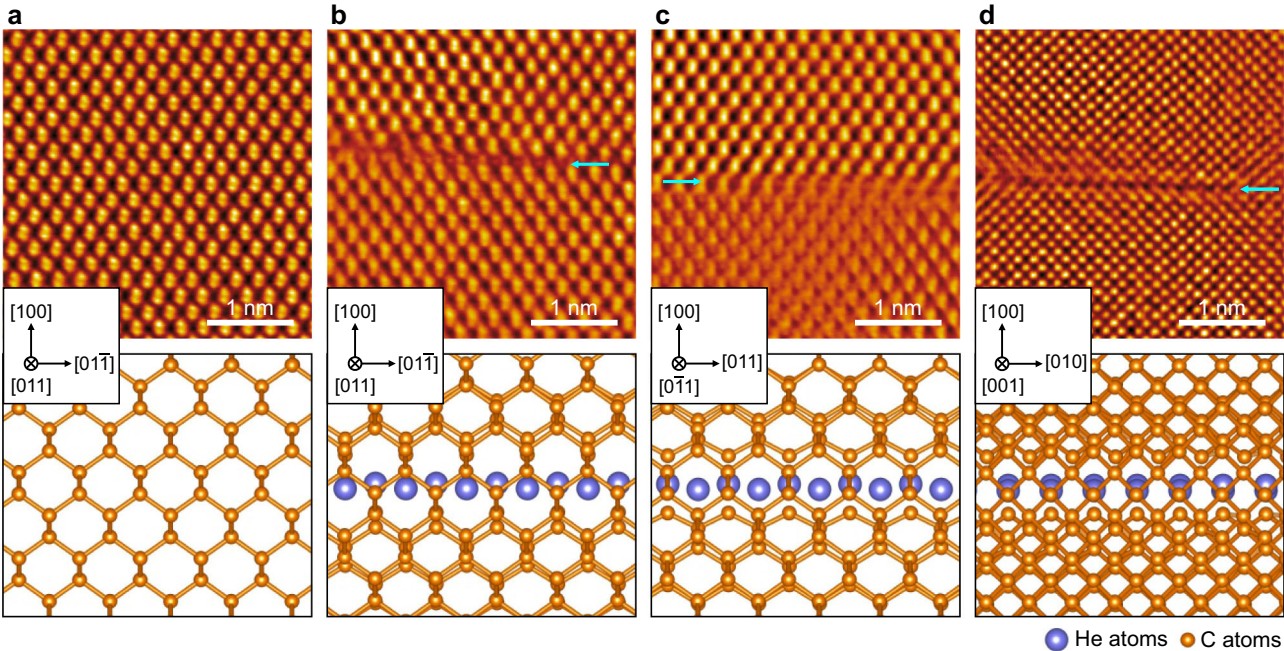

**Fig. 3 | Evidence of two-dimensional solid helium confined by diamond. a** iDPC STEM images (upper) and the corresponding simulated atomic structure (lower) of the pristine diamond viewed along the [011] zone axis. iDPC STEM images (upper) and the corresponding simulated atomic structures (lower) of diamonds containing two-dimensional solid helium viewed from (**b**) [011], (**c**) [0$\bar{1}$1] and (**d**) [001] zone axes. Blue arrows highlight helium platelets confined by the diamond lattice. The top and bottom diamond lattices are included in the simulated atomic structures.

decreasing the platelet spacing. Moreover, we measured the spacings between helium platelets by iDPC STEM imaging (see Supplementary Fig. 7c). The average spacing between (100) helium platelets is 3.0 ± 1.3 nm, and the minimum platelet spacing is about 1 nm, corresponding to the maximum helium pressure of about 166 GPa. Such a strong lattice confinement effect, which cannot be produced in conventional materials, enables us to achieve the two-dimensional solid helium successfully. In light of experimental evidence, we considered the geometrical matching of helium atoms to the surrounding diamond lattice and proposed an atomic structural model of two-dimensional solid helium (see Methods). Figure 2c depicts the atomic structure relaxed by first-principles DFT calculations (see cif file in Supplementary Data 1). The accurate atomic positions are shown in Fig. 2d, and two-dimensional solid helium has an areal density of 0.315 atom/Å$^2$.

We further demonstrate the anisotropic atomic arrangement within {100} helium platelets, which has a qualitative match to the atomic structural model viewed from different directions. Figure 3a shows the representative iDPC STEM image of diamond lattice beneath the helium region along the [011] zone axis. By contrast, two-dimensional solid helium with a buckled tetragonal structure (highlighted by the blue arrow) is displayed in Fig. 3b when the [011] zone axis was adopted. Moreover, the iDPC STEM image collected under the [0$\bar{1}$1] zone axis (see Fig. 3c) reveals the anisotropic nature of the two-dimensional solid helium. These findings are well supported by the simulation of iDPC STEM images under the <011> directions (see Methods and Supplementary Fig. 8). In addition, no additional helium atoms (see Fig. 3d) could be detected from these results under the [001] zone axis due to the overlap of helium and carbon atoms. Apparently, this kind of two-dimensional solid helium differs drastically from its three-dimensional counterparts, which have hexagonal close-packed or face-cantered cubic structures[11].

Figure 4 further summarizes the pressure-temperature (P-T) domain for solid helium in different dimensions. High-pressure techniques enabled the discovery of three-dimensional solid helium at high P-T conditions[8–11,37] but previously remained challenging in stabilizing two-dimensional solid helium. Observations of two-dimensional solid helium were made with adsorption on graphite at low temperatures (between 1.2 and 7.4 K)[17–19]. With the addition of the pressure dimension, we achieved two-dimensional solid helium at ambient temperature. Our unprecedented findings will boost the scientific discovery of monolayer solid helium in the future. More importantly, distinguishing from the conventional diamond anvil cell used for high-pressure research, the utilization of the robust diamond lattice can render us a clear visualization of its atomic structure and realize the controllable manipulation of them to achieve various unusual properties at ambient temperature.

## Strain doping

We found that these high-pressure two-dimensional solid helium atoms can generate considerable elastic strain (see strain mapping in Supplementary Fig. 9a–d) to the diamond lattice. Such a distinctive strain doping effect allows us to substantially change the intrinsic electronic features, resulting in unique applications of them in semiconductor electronics. For instance, we used the off-axis STEM-EELS to measure the bandgap of the helium region and the pristine diamond by enforcing a specific momentum transition selection (see Methods). The results suggested that compressive strains induced by two-dimensional solid helium could give rise to the bandgap narrowing of diamond as much as 2.2 eV (see Supplementary Fig. 9e). Such a pronounced change of band structures of the <100>-oriented diamond lattice subjected to different compressive strains was further confirmed by DFT calculations (see Supplementary Fig. 9f). Benefiting from the two-dimensional solid helium, deep elastic strain engineering of diamond is expected to be reliably achieved. Previous reversible strain engineering is mainly based on nanomechanical techniques[38,39], which are only appropriate for small-volume diamond nanostructures. By contrast, our strategy paves a way for reserving large lattice strains in bulk diamond by introducing intrinsic defects, which should be applicable to other covalent semiconductors with high elastic strength. The ion implantation process may be readily applied to diamond wafers in the existing semiconductor factory. By further

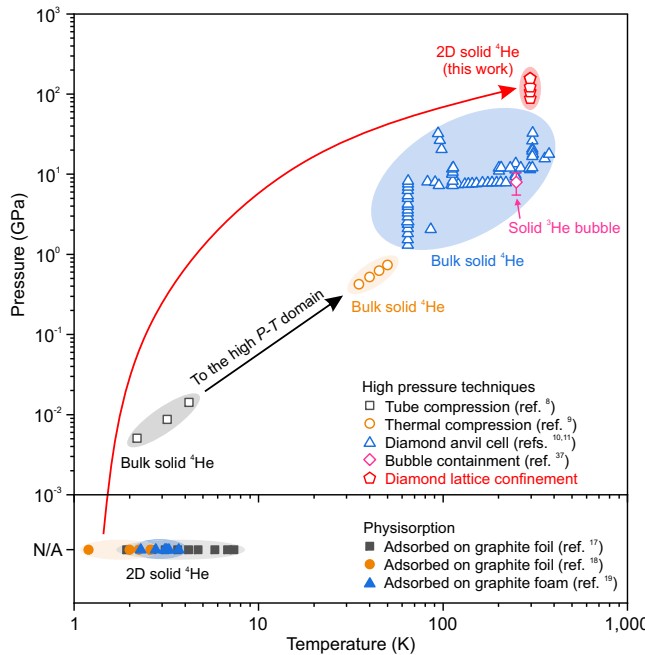

**Fig. 4 | P-T domain of three-and two-dimensional solid helium.** Helium was first solidified by Keesom[8] in 1926. With the development of high-pressure techniques[9–11,37], three-dimensional solid helium has been obtained at high P-T conditions. Nevertheless, two-dimensional solid helium has previously been achieved by physiorption on the surface of graphite at low temperatures[17–19]. Using diamond lattice confinement, we substantially extended the P-T domain of two-dimensional solid helium for potential practical applications. Source data are provided as a Source Data file.

adopting masking techniques, which have been maturely used in semiconductor manufacturing, the controllable distributions of two-dimensional solid helium would be efficiently realized for achieving tunable properties.

## Discussion

Diamond with the $sp^3$ bonding configuration is metastable as compared to graphite with stable $sp^2$ bonds. Therefore, ion implantation can break some $sp^3$ bonds of diamond, resulting in the formation of amorphous carbon and graphite. When irradiation-induced vacancy concentration exceeds a critical threshold of damage $D_c$, ranging from $1 \times 10^{22}$ to $9 \times 10^{22}$ vacancies cm$^{-3}$, an amorphous phase of diamond will be produced at room temperature[30,31,40]. The damaged diamond can further transform to graphite or transform back to diamond at high temperatures[41–43], dependent on the conditions for the re-arrangement of broken bonds. Generally speaking, low-dose implantation can be annealed by recrystallization, whereas high-dose implantation results in graphitization. Here, suppression of irradiation-induced amorphization and graphitization of diamond is the prerequisite for the formation of two-dimensional solid helium. According to the result of SRIM calculation, helium dose in the present condition introduced damage of up to $1.7 \times 10^{23}$ vacancies cm$^{-3}$ into diamond, which is much higher than the critical threshold, i.e., $D_c = 1–9 \times 10^{22}$ vacancies cm$^{-3}$. However, the irradiation-induced instability in diamond is also influenced by temperature. We have conducted more than ten batches of implantation experiments with different temperatures ranging from room temperature to 1573 K. Our TEM observations demonstrated that helium ion implantation had little influence on the diamond structure at temperatures higher than 1173 K (see Supplementary Fig. 10). It is because high temperature can enhance the point defect recombination rate that mitigates the damage production. In general, amorphization can become more difficult in semiconductors and ceramics with

increasing temperature[44,45]. The relationship between the temperature $T$ and ion dose $D$ to achieve the amorphous state is expressed as[44]:

$$\ln(1 - D_0/D) = C - \frac{E_{act}}{kT} \qquad (1)$$

where $D_0$ is the amorphization dose at low temperature, $C$ is a constant dependent on the ion flux and damage cross-section[46], $E_{act}$ is the activation energy, and $k$ is the Boltzmann constant. Therefore, the ion dose $D$ for triggering of amorphization and graphitization in diamond will increase as the temperature $T$ increases. However, the high-temperature transformation of diamond to graphite will occur at temperatures higher than ~1873 K[47]. As a result, in this work, we selected an implantation temperature of 1573 K to simultaneously suppress irradiation and thermal-induced graphitization in diamond, as demonstrated by electron diffraction (see Fig. 1d) and STEM-EELS measurement (see Fig. 1g). Moreover, such a high temperature can result in the formation of helium platelets with a larger diameter, which is conducive to high-quality S/TEM observations (see Fig. 2a, b).

Helium will accumulate in crystalline materials due to tritium decay, direct α-injection, or ($n$, α) transmutation reactions[27]. Since helium solubility in solids is limited[20], helium atoms are strongly inclined to form two types of helium-filled nanoscale cavities: (1) helium nanobubbles and (2) helium platelets. Helium platelet, where helium atoms are confined in a two-dimensional structure, is a special nano-bubble in crystalline materials. Helium platelets are usually formed in the case of negligible availability of vacancies. It is energetically more favorable for helium interstitial atoms to form a two-dimensional structure between two atomic layers of matrix lattices. In other words, no matrix atom diffusion is required in the formation of helium plate-lets, which are systems far from equilibrium. The underlying reason for the {001} habit plane of helium platelets could be the contribution of elastic free energy to the total energy of helium platelets. Diamond is the most incompressible substance in nature. The <100> direction is elastically softest in diamond[39], resulting in the smallest elastically stored energy. Therefore, the formation of {001} helium platelets is thermodynamically favored in diamond. The same {001} habit plane has been also reported in helium-implanted silicon[48,49], which has the same crystal structure and similar elastic anisotropy with diamond. Due to the elastic anisotropy of diamond, our solid helium platelets results are inconsistent with the previous reports of solid neon[50] and xenon[51] platelets in metals, which are parallel to close-packed planes of the matrix.

An important aspect in helium investigations was the determination of helium density and pressure in the nanobubbles. The choice of a suitable equation of state (EOS) to correlate density and pressure was discussed as a function of temperature and helium density range[52]. The EOS given by Trinkaus[53] was considered the most accurate for nano-bubbles in the density range up to a hundred atom/nm$^3$. In this contest EELS at the He $K$-edge has been used to quantify helium density inside the nanobubbles. Starting from 21.218 eV for the free helium atoms[54], the energy of the 1s → 2p transition was blue shifted by the decrease of pore size associated with increased helium density and pressure for different matrix elements[21–23]. This phenomenon has been attributed to a short-range Pauli repulsion between the electrons of neighboring atoms[55,56]. As expected, low-voltage monochromatic STEM-EELS mea-surements (see Fig. 1f) demonstrate the blue shifted He $K$-edge at ~25.0 eV from the helium platelet, corresponding to a blue shift of ~3.8 eV. We then compared the blue shift with theoretical predictions at ultra-high pressure. Using self-consistent electronic structure cal-culations, Jäger et al.[57] derived a linear relation between the energy shift $\Delta E$ and helium density $n_{He}$ of atoms per unit volume of solid face-centered cubic (fcc) helium:

$$\triangle E \text{ (eV)} = \chi + C_n n_{He} = -0.8 + 22n \left( \text{Å}^{-3} \right) \qquad (2)$$

where the intercept $\chi$ and the gradient $C_n$ are two fitting parameters. To consider the two-dimensional system, the density $n_{He}$ is related to the closest inter-atomic separation $R$ through $n_{He} = \sqrt{2}/R^3$. The energy shift in (2) can be rewritten in terms of the inter-atomic distance $R$:

$$\triangle E\,(\text{eV}) = -0.8 + 31.1R^{-3}(\text{Å}) \qquad (3)$$

With $R = 1.8$ Å given by iDPC STEM image and DFT calculations, Eq. (3) yields $\Delta E = 4.5$ eV. The energy shift $\Delta E$ measured by EELS is reasonably smaller than the theoretical value because the $2p$ orbital of a fourfold coordinated atom in the two-dimensional structure will experience a weaker compressive effect than a helium atom having a larger number (12) of closest neighbors in solid fcc helium.

## Methods

### Helium ion implantation
Single-crystal type Ib <100> diamonds (HSCD13A) were produced by Henan Huanghe Whirlwind Co. Ltd., China. The nitrogen content is less than 100 ppm. The mirror-polished diamond samples have dimensions of 4 mm × 4 mm × 1 mm. Diamonds were implanted with 275 keV $^4$He$^+$ ions at 1573 K using High Voltage Engineering Europe 500 kV ion implanter at the Accelerator Laboratory, National Tsing Hua University. The beam flux was maintained at $1.2 \times 10^{13}$ ions cm$^{-2}$ s$^{-1}$, and the total fluence was $6.4 \times 10^{16}$ ions cm$^{-2}$. During implantation, diamonds were placed in a vacuum chamber with pressure below $10^{-6}$ torr, maintained by an oil rotary pump and a turbomolecular pump, and heated by a molybdenum (Mo) heater (see Supplementary Fig. 1a). The curved bottom of the Mo sheet can tightly clamp diamond samples at high temperatures. The sample temperature was precisely controlled (±5 K) based on active temperature feedback from platinum/platinum-rhodium (Pt/Pt-Rh) thermocouples. A hole with a diameter of 1.0 mm was machined from the Mo heater to measure ion fluence by a Faraday cup. Irradiation damage and helium concentration along the cross-section direction (see Supplementary Fig. 1b) were predicted by the stopping and range of ions in matter (SRIM-2013) code[58] with the quick Kinchin-Pease mode. The displacement threshold energy was chosen as 37.5 eV for the <100> diamond[59], and the lattice binding energy was zero. The diamond density used for the calculation was 3.514 g cm$^{-3}$.

### Specimen preparation
After the $^4$He$^+$ ion implantation, diamonds were sputtered with conductive silver thin film by a Denton Vacuum Desk V sample preparation system to prevent charging during scanning electron microscopy (SEM) imaging and FIB processing. The cross-sectional TEM specimens were prepared by FIB lift-out techniques (see Supplementary Fig. 2a–f) using the FEI Scios DualBeam SEM/FIB system. Prior to the FIB machining, electron beam-induced Pt deposition was performed on the surface of diamonds to prevent damage. Subsequently, the diamond lamella was mounted to an Omniprobe lift-out copper grid and thinned to electron transparency. During this stage, the accelerating voltage of gallium (Ga$^+$) ion beams was progressively decreased from 30 to 5 kV. Finally, Ga$^+$ ion beams of 2 kV per 4.3 pA were employed to minimize the amorphous damaged layers. An example of a high-quality TEM specimen with only a 3-nm-thick amorphous layer is shown in Supplementary Fig. 2g–i.

### Microstructural characterization
A JEOL JEM-2100F TEM operated at 200 kV was used to perform Fresnel contrast imaging, selected area electron diffraction, and high-resolution TEM imaging of diamond specimens. HAADF STEM imaging of helium platelets was conducted using a JEOL JEM-ARM200F aberration-corrected STEM operated at 200 kV with a detector acceptance semi-angle of 40–160 mrad and a probe current around 60 pA. Atomic-resolution iDPC STEM imaging was conducted using an aberration-corrected Thermo Fisher Themis Z STEM operated at 300 kV. The

thickness of thin regions of the diamond specimens for iDPC STEM experiments is about 20 nm. All iDPC STEM images were acquired with a probe convergence semi-angle of 21 mrad and a probe current around 30 pA. The microscope has DPC1-4 detectors with collection angles of 7–29 mrad (see Supplementary Fig. 4a) for iDPC STEM imaging and an ADF detector with a collection angle of 31–186 mrad for geometric phase analysis. The experimental iDPC STEM images were filtered by applying an average background subtraction filter and a high-pass Gaussian filter without changing the atomic structure according to the procedure of de Graaf et al.[35].

### Atomic structural model
The helium layers were arranged in the middle of the supercell to screen out the pseudo-interactions between periodically repeated layers. The initial crystal structure of two-dimensional solid helium was constructed by inserting helium atomic layers into the cubic diamond. The lattice constant $a$ of the pristine diamond was predicted to be 3.563 Å, showing good agreement with the experimental value of 3.567 Å. In particular, a supercell was built up from the equilibrium diamond cell with lattice vectors defined as [110], [$\bar{1}$10], and [004] and thus consists of 16 (001) carbon atomic layers. Three different sites of interstitial helium in the diamond lattice (tetrahedral sites $T$, hexagonal sites $H$, and bond-centered sites $BC$) and two different in-plane atomic ratios (He:C = 1:1 and 2:1) were considered and calculated. According to the relaxed atomic models, the possible configuration of two-dimensional solid helium was illustrated in Fig. 2c, d. Besides, configurations with 1 helium layer arranged in the middle of 12, 20, and 24 (001) carbon atomic layers were constructed to investigate the effect of helium interlayer spacing on cell pressure at the limits of computing. To modulate helium platelet confined in diamond lattice, all atomic positions and lattice parameters were fully relaxed only for the initial diamond by DFT calculations. Optimization on atomic internal freedoms was performed for the other structural models. The structural models were visualized using the VESTA program[60].

### Ab initio calculations
DFT calculations were performed using the projector augmented wave method[61] as implemented in the Vienna Ab initio Simulation Package[62]. The exchange-correlation functional was described by the generalized gradient approximation with the Perdew-Burke-Ernzerhof formulation[63]. Configurations of $2s^2 2p^2$ for carbon and $1s^2$ for helium were employed as valence electrons. The electronic Bloch wave function was expanded using a plane-wave basis set with an energy cutoff of 650 eV. The convergence criteria of electronic and ionic optimizations were set to be $10^{-7}$ eV for total energy and 0.001 eV/Å for Hellmann-Feynman force, respectively. The electronic Brillouin zones were sampled by Monkhorst-Pack $k$-meshes[64] with a resolution of $2\pi \times 0.025$ Å$^{-1}$. All the calculations were performed with spin polarization. The van der Waals interaction was considered using the DFT-D3 dispersion correction method[64].

### STEM-EELS
EELS analysis was performed by using the FEI Titan Cubed Themis G2 300 STEM equipped with a Gatan GIF Quantum ERS spectrometer. The microscope was operated at 60 kV and monochromated with an energy resolution of 0.30 eV, determined by the full-width at half-maximum of the zero-loss peak. The low-loss EELS data on and off the platelet were acquired with a convergence semi-angle of 25 mrad, a collection angle of 22.5 mrad, and an energy dispersion of 0.01 eV per channel. For measuring the carbon $K$ edge of diamond, the core-loss EELS data were acquired with a convergence semi-angle of 25 mrad, a collection angle of 22.5 mrad, and an energy dispersion of 0.1 eV per channel from an area of ~30,000 nm$^2$ with a step size of 50 nm. Off-axis STEM-EELS, which can remove Čerenkov loss and surface effects to reveal the exact interband transition onset[65,66], was used to investigate

the effects of two-dimensional solid helium on the bandgap of diamond. For obtaining the separated convergent beam electron diffraction disks, the convergence angle and collection angle were reduced to 8.6 mrad and 5.75 mrad, respectively. Then, the $1\bar{1}1$ Bragg diffraction disk with a high **q** value in momentum space was deliberately selected to enter the EELS aperture via beam shift. EELS data were acquired with a dispersion of 0.01 eV per channel and an exposure time of 1 s.

### iDPC STEM image simulations

At the limits of multislice computing, we built an atomic model of 6.05 nm × 6.05 nm × 2.14 nm in dimensions (see Supplementary Fig. 8a), containing 13,824 carbon atoms and 578 helium atoms. Multislice simulations were conducted using the Dr. Probe GUI program[67] (Version 1.9). Atomic vibrations were considered in simulations by the frozen-lattice method, where the Debye-Waller factors used at 300 K are 0.14 $\text{Å}^2$ for C (diamond structure)[68] and 0.1 $\text{Å}^2$ for solid $^4$He (ref. [69]), respectively. The simulated microscope was set to have no aberrations, and the probe conditions were set equal to the experimental settings of 300 kV with a convergence angle of 21 mrad. The probe step size was set to 10 pm. The simulated images from the DPC1-4 detectors with an acceptance semi-angle of 7–29 mrad were then processed to obtain the iDPC image, according to the method described in the work of Lazic et al.[34]. The opposite segments were subtracted to obtain the orthogonal (x,y) DPC images $\text{DPC}_x$ and $\text{DPC}_y$, which were then integrated into the Fourier domain to obtain the simulated iDPC STEM image.

### Geometric phase analysis

Lattice strain mapping was measured using the open-source Strain++ program (Version 1.7) according to the geometric phase analysis algorithm[70]. In practice, the fast Fourier transform patterns were obtained from the atomic-resolution ADF STEM images, and two {111} Bragg reflections were chosen for obtaining the strain mapping of the close-packed planes (see Supplementary Fig. 9b). Finally, the horizontal normal strain ($\varepsilon_{xx}$) and vertical normal strain ($\varepsilon_{yy}$) maps are shown in Supplementary Fig. 9c, d, respectively. It demonstrates that pressurized helium platelets generate compressive strains to the surrounding diamond lattice.

### Reporting summary

Further information on experimental design is available in the Nature Research Reporting Summary linked to this paper.

## Data availability

The authors declare that all data supporting the findings of this study are available within the paper and its Supplementary Information files. Source data are provided with this paper.

## Code availability

The codes used for DFT calculations and iDPC STEM image simulations in this study are available from the corresponding authors upon request.

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

## Acknowledgements

We thank the director of Accelerator Laboratory, Dr. H. Niu, at National Tsing Hua University for giving us access to the $^4$He$^+$ ion beamline and Mr. H.K. You for the technical support. We are grateful to Prof. A. Kouchi and Dr. T. Yamazaki at Hokkaido University for performing in situ cooling TEM experiments to explore solid helium inside nanobubbles at the initial stage of this project. J.J.K. was supported by the Hong Kong Research Grants Council (Nos. CityU11209021, CityU11214820, and CityU11205018). T.Y. was financially supported by the Hong Kong Research Grants Council (No. CityU21205621). Y.L. acknowledges the funding support from the National Natural Science Foundation of China (No. 11922215) and the Research Grants Council of the Hong Kong Special Administrative Region, China (No. RFS2021-1S05). J.L. and G.W. acknowledged the support from the National Natural Science Foundation of China (No. 11974156), Guangdong Innovative and Entrepreneurial Research Team Program (Grant No. 2019ZT08C044), Shenzhen Science and Technology Program (No. KQTD20190929173815000), and the technical support from SUSTech Core Research Facilities and Pico-Centre.

## Author contributions

W.L. and J.J.K. conceived the research. W.L., S.L., D.C. and C.H.Y. performed high-temperature $^4$He$^+$ ion implantation. W.L. carried out SRIM calculations, FIB microfabrication and TEM observations. G.W. and J.L. conducted STEM-EELS characterization. S.d.G. and B.J.K. performed

iDPC STEM imaging. Y.R.L. proposed the atomic structure of two-dimensional solid helium. Y.R.L., J.F., and S.Z. performed DFT calculations and built atomic structural models. W.L. and H.Z. performed iDPC STEM image simulations. W.L., Y.R.L., G.W., T.Y., Y.L. and C.T.L. wrote the manuscript. All authors discussed the results and contributed to the final manuscript.

## Competing interests

The authors declare no competing interests.
