## [Peer Review File · Nature Communications]

REVIEWER COMMENTS

Reviewer #1 (Remarks to the Author):

The paper is of high interest for the fabrication of 2D solid-Helium by confinement between planes in a diamond lattice. Especially at the level of TEM instruments of last generation with most advanced techniques incorporated. Experimental results are also combined with modelling and simulations.

However references to previous works on Helium bubbles and platelets stabilized at room temperature in different solid matrices after ion implantations are limited. The concept of Helium trapped in a "condensed state" at room temperature has been widely investigated for Helium implanted metals mainly in relation to damage in materials for nuclear reactors (1). The introduction of Helium bubbles (cavities) found applications in Silicon technology and Helium trapped in Silicon has been also widely investigated (2-5).

An important aspect in these investigations was the determination of He density and pressure in the nano-bubbles. The choice of a suitable equation of state (EOS) to correlate density and pressure was discussed as a function of temperature and He density range (6). The Trinkaus EOS (7) was also considered the most accurate for nanobubbles in the density range up to a hundred at/nm³. In this context EELS at the He K-edge has been used to quantify He density inside the nanobubbles. Starting from 21.2 eV for the free Helium, the energy of the 1s→2p transition was blue shifted by pore size decrease associated to increased density and pressure for different matrix elements (2-4). This phenomenon has been attributed to a short-range Pauli repulsion between the electrons of neighboring atoms (8-9).

It will be of interest if the authors can discuss the EELS He K-edge results considering the given ultra-high pressure value of 56 GPa in their diamond matrix.

(1) S.E. Donnelly, J.H. Evans (eds). "Fundamental Aspects of Inert Gases in Solids", Springer Science+Business Media New York © 1991. Originally published by Plenum Press, New York in 1991. [doi: 10.1007/978-1-4899-3680-6]

(2) K. Alix, M.-Laure David, G. Lucas, D.T.L. Alexander, F. Pailloux, C. Hébert, L. Pizzagalli. Gentle quantitative measurement of helium density in nanobubbles in silicon by spectrum imaging. *Micron* 77 (2015) 57–65. [<http://dx.doi.org/10.1016/j.micron.2015.05.011>]

(3) R. Schierholz, B. Lacroix, V. Godinho, J. Caballero-Hernández, M. Duchamp, A. Fernández, STEM–EELS analysis reveals stable high density He in nanopores of amorphous silicon coatings deposited by magnetron sputtering, *Nanotechnology* 26 (2015) 075703. [doi:10.1088/0957-4484/26/7/075703]

(4) B. Lacroix, V. Godinho, A. Fernández, The nanostructure of porous cobalt coatings deposited by magnetron sputtering in helium atmosphere, *Micron* 108 (2018) 49–54. [doi: 10.1016/j.micron.2018.02.004]

(5) X. Hebras, P. Nguyen, K.K. Bourdelle, F. Letertre, N. Cherkashin , A. Claverie, Comparison of platelet formation in hydrogen and helium-implanted silicon, Nuclear Instruments and Methods in Physics Research B 262 (2007) 24–28. [doi:10.1016/j.nimb.2007.04.158]

(6) C.A. Walsh, J. Yuan, L.M. Brown. M.,. A procedure for measuring the helium density and pressure in nanometre-sized bubbles in irradiated materials using electron-energy-loss spectroscopy. Philos. Mag. A 80 (2000) 1507–1543. <http://dx.doi.org/10.1080/01418610008212134>.

(7) Trinkaus, H., 1983. Energetics and formation kinetics of helium bubbles in metals. Radiat. Eff. 78, 189–211. <http://dx.doi.org/10.1080/00337578308207371>.

Trinkaus, H., 1983. Energetics and formation kinetics of helium bubbles in metals. Radiat. Eff. 78, 189–211. <http://dx.doi.org/10.1080/00337578308207371>.

(8) A.A. Lucas, J.P. Vigneron, S.E. Donnelly, J.C. Rife. Theoretical interpretation of the vacuum ultraviolet reflectance of liquid helium and of the absorption spectra of helium microbubbles in aluminum Phys. Rev. B. 28 (1983) 2485.

(9) N. C. Pyper, T.C. Naginey, C.T. Whelan. Environmental modifications of atomic properties: The ground and 1s2p excited states of compressed helium. J Chem. Phys. 155 (2021) 214301. [doi: 10.1063/5.0066626]

Asunción Fernández

Instituto de Ciencia de Materiales de Sevilla, Spain

Reviewer #2 (Remarks to the Author):

This manuscript contains some Interesting novel experimental observations. In particular, the evidence for formation of solid platelets of He in implanted diamond and the accompanying reduction in the diamond bandage by >2 eV are important observations. However, the scientific interpretation and discussion of the results should be strengthened. Several specific comments are listed below.

Intro, p. 3. Why did the authors select a relatively extreme implantation temperature of 1573K? My understanding is that amorphization and graphitization can be suppressed in ion irradiated diamond at implantation temperatures of ~1370K or lower. Some discussion of the temperature-dependent stability of diamond vs graphite should be provided here (and on p. 4).

Larger font sizes should be used for axes labels and text in Fig. 1, 2, 3 and Supplementary Fig. 1b. (also typo in Suppl. Fig. 1b y axis label should be fixed; vacancies/cm⁻³ should be vacancies/cm³).

Bottom p. 3: Why was a $\langle 001 \rangle$ oriented diamond crystal considered preferable to other orientations ($\langle 111 \rangle$, etc.)? On p 4, it is noted that 001 platelets also formed in polycrystalline diamond for a $\langle 311 \rangle$ oriented grain. (relatively close to 001, vs. 110 or 111 orientation).

Based on prior studies on He-implanted ceramics such as silicon, SiC and Al₂O₃, platelets typically form on the low index plane closest to the surface orientation, and pronounced orientation effects for He-implanted diamond have been reported by other research groups (see M. Chen, et al., ref. 23; S.J. Zinkle, Nucl. Instrum. Meth. B 286 (2012) 4-19; and many other journal articles). Have the authors studied the effect of orientation (with respect to the irradiated surface) on formation of 2D platelets in diamond? Figs. 1b, 1c seems to show that platelets are located on 001 plane aligned with surface, whereas Fig. 1e shows some platelets on two perpendicular $\{001\}$ habit planes; what was the relative fraction of platelets on the three $\{001\}$ habit planes? (strongly anisotropic vs. nearly isotropic distribution?). Streaks in the SAED in Fig. 1d seem to be visible in only one $\{001\}$ habit plane, suggesting that there was a strong anisotropy in the preferred habit plane. What was average platelet diameter?

p. 5: what was typical spacing between 001 platelets? Is there any physical significance to this spacing? What controls the partitioning of the He into platelets? Is there a proposed reason why the He platelets reported by J. Chen et al. (1999) in SiC had a platelet width of $\sim 2-3$ atomic spacings, whereas your He-implanted diamond consisted of only one atomic layer?

top of p. 6. How was 56 GPa lattice pressure estimated? (this is not clear from the text in the methods section and Suppl. Fig. 7)

top of p. 7. Are results consistent with previous reports of solid Ne, Ar, Kr platelets in implanted ceramics? (monolayer vs. multi-atomic thickness platelets). E.g., E.L. Fleischer, M.G. Norton. Noble gas inclusions in materials, Heterogeneous Chemistry Reviews 3 (1996) 171-201.

p. 7 Does the formation of He platelets produce compressive vs. tensile strains? Since the free surface can expand, large compressive lattice strain would not be expected in the matrix, whereas tensile strains would be expected to exist at the periphery of the platelets.

Suppl. Fig. 7d seems to show high compressive strains within the platelets, and high tensile strains in the diamond lattice.

p. 8, last sentence: rewording needed. "achieving well-target and extraordinary properties"

Reviewer #3 (Remarks to the Author):

In this work, two-dimensional solid helium is stabilized at room temperature within a diamond lattice matrix by means of ion-implantation techniques. The structure of the helium-based layers are characterized by experimental spectroscopy and computational DFT techniques.

Despite that the technical quality of the present work may acceptable, I find that the present work does not advance any significant knowledge in the fundamental or applied sciences, as I explain in

more detail next. In addition to such a lack of general interest and scientific relevance, I find that the authors fail in their attempt to assess the current state-of-the-art in the field of low-dimensional solid helium (or quantum solids). Therefore, I do not recommend publication of the present work in Nature Communications.

As a matter of fact, two-dimensional solid helium has already been realized on top of common substrates like graphite [see, for instance, Journal of the Physical Society of Japan Vol. 77, No. 11, November, 2008, 111013 "Nuclear Magnetism in Two-Dimensional Solid Helium Three on Graphite" and Nature Physics volume 13, pages 455–459 (2017) "Intertwined superfluid and density wave order in two-dimensional 4He "] and carbon nanotubes [see, for instance, EPL, 88 (2009) 56005 "Dynamics of one-dimensional and two-dimensional helium adsorbed on carbon nanotubes"]. Therefore, there is not any novelty in stabilizing low-dimensional solid helium.

The interest of studying two-dimensional solid helium at low temperatures ($T \sim 1\text{K}$) is purely fundamental: helium is the archetypal quantum crystal and lowering its dimensionality may lead to new quantum atomic physics and exotic states of matter (e.g., helium atoms are boson particles and can form Bose-Einstein condensates). However, there is not any fundamental interest in studying low-dimensional solid helium at room temperature and high pressures since under those thermodynamic conditions the quantum nature of solid helium is totally absent and the system behaves classically (i.e., no atomic delocalization, no Bose-Einstein condensation, no exotic states of matter can emerge -see, for instance, Rev. Mod. Phys. 89, 035003 (2007) "Simulation and understanding of atomic and molecular quantum crystals"). On the other hand, I fail to appreciate any practical interest and relevance out of synthesizing two-dimensional solid helium in a diamond matrix; on this regard, the authors mention some possible "intrinsic train doping" but that is something that could be equally achieved by using other atomic light species (e.g., Li) and same ion implantation techniques (...probably even more easily than with helium!).

In addition to these lacks of novelty and scientific and practical interests, I find that some of the claims made by the authors in their manuscript are not sufficiently well justified. For instance, in the abstract it appears written that solid helium monolayers were observed at "ultra-high pressures up to 56 GPa". This information gives the impression to the readers that the authors have performed some high-pressure diamond anvil cell experiments (DAC) under well-defined hydrostatic pressure conditions. However, one rapidly realizes in reading the subsequent sections that such a claim is quite misleading since it is based on a very crude elastic approximation (explained in Sec. "Helium pressure imposed by the diamond lattice") that completely disregards any atomistic detail. [...Actually, how a ultrahigh pressure of 56 GPa can be maintained in a stable manner in the interior of a matrix that as a whole is in equilibrium at ambient pressure?? To most scientists versed on thermodynamics this statement does not make any physical sense...]. Moreover, the exact crystal structure of the stabilized two-dimensional solid helium is not satisfactorily characterized (i.e., the corresponding cif file appears to be missing).

For all these reasons, unfortunately, I cannot be more positive in my evaluation.

Response to the Reviewers

We deeply appreciate the reviewers' constructive comments and valuable suggestions, which help us greatly improve the quality and presentation of our research work. Based on these comments, we have carefully revised the manuscript. Below is a point-by-point response to each comment and suggestion.

Response to Reviewer #1:

Comment 1: *"The paper is of high interest for the fabrication of 2D solid-Helium by confinement between planes in a diamond lattice. Especially at the level of TEM instruments of last generation with most advanced techniques incorporated. Experimental results are also combined with modelling and simulations.*

Reply 1: Thanks a lot for the positive comment on our work.

Comment 2: *"However references to previous works on Helium bubbles and platelets stabilized at room temperature in different solid matrices after ion implantations are limited. The concept of Helium trapped in a "condensed state" at room temperature has been widely investigated for Helium implanted metals mainly in relation to damage in materials for nuclear reactors (1). The introduction of Helium bubbles (cavities) found applications in Silicon technology and Helium trapped in Silicon has been also widely investigated (2-5). An important aspect in these investigations was the determination of He density and pressure in the nano-bubbles. The choice of a suitable equation of state (EOS) to correlate density and pressure was discussed as a function of temperature and He density range (6). The Trinkaus EOS (7) was also considered the most accurate for nanobubbles in the density range up to a hundred at/nm³. In this contest EELS at the He K-edge has been used to quantify He density inside the nanobubbles. Starting from 21.2 eV for the free Helium, the energy of the 1s→2p transition was blue shifted by pore size decrease associated to increased density and pressure for different matrix elements (2-4). This phenomenon has been attributed to a short-range Pauli repulsion between the electrons of neighboring atoms (8-9).*

It will be of interest if the authors can discuss the EELS He K-edge results considering the given ultra-high pressure value of 56 GPa on their diamond matrix."

(1) S.E. Donnelly, J.H. Evans (eds). "Fundamental Aspects of Inert Gases in Solids", Springer Science+Business Media New York © 1991. Originally published by Plenum Press, New York in 1991. [doi: 10.1007/978-1-4899-3680-6]

(2) K. Alix, M.-Laure David, G. Lucas, D.T.L. Alexander, F. Pailloux, C. Hébert, L. Pizzagalli. Gentle quantitative measurement of helium density in nanobubbles in silicon by spectrum imaging. *Micron* 77 (2015) 57–65. [http://dx.doi.org/10.1016/j.micron.2015.05.011]

(3) R. Schierholz, B. Lacroix, V. Godinho, J. Caballero-Hernández, M. Duchamp, A. Fernández, STEM–EELS analysis reveals stable high density He in nanopores of amorphous silicon coatings deposited by magnetron sputtering, *Nanotechnology* 26

(2015) 075703. [doi:10.1088/0957-4484/26/7/075703]

(4) B. Lacroix, V. Godinho, A. Fernández, *The nanostructure of porous cobalt coatings deposited by magnetron sputtering in helium atmosphere*, *Micron* 108 (2018) 49–54. [doi: 10.1016/j.micron.2018.02.004]

(5) X. Hebras, P. Nguyen, K.K. Bourdelle, F. Letertre, N. Cherkashin, A. Claverie, *Comparison of platelet formation in hydrogen and helium-implanted silicon*, *Nuclear Instruments and Methods in Physics Research B* 262 (2007) 24–28. [doi:10.1016/j.nimb.2007.04.158]

(6) C.A. Walsh, J. Yuan, L.M. Brown. M.,. *A procedure for measuring the helium density and pressure in nanometre-sized bubbles in irradiated materials using electron-energy-loss spectroscopy*. *Philos. Mag. A* 80 (2000) 1507–1543. <http://dx.doi.org/10.1080/01418610008212134>.

(7) Trinkaus, H., 1983. *Energetics and formation kinetics of helium bubbles in metals*. *Radiat. Eff.* 78, 189–211. <http://dx.doi.org/10.1080/00337578308207371>.

Trinkaus, H., 1983. *Energetics and formation kinetics of helium bubbles in metals*. *Radiat. Eff.* 78, 189–211. <http://dx.doi.org/10.1080/00337578308207371>.

(8) A.A. Lucas, J.P. Vigneron, S.E. Donnelly, J.C. Rife. *Theoretical interpretation of the vacuum ultraviolet reflectance of liquid helium and of the absorption spectra of helium microbubbles in aluminum* *Phys. Rev. B.* 28 (1983) 2485.

(9) N. C. Pyper, T.C. Naginey, C.T. Whelan. *Environmental modifications of atomic properties: The ground and 1s2p excited states of compressed helium*. *J Chem. Phys.* 155 (2021) 214301. [doi: 10.1063/5.0066626]

Asunción Fernández

Instituto de Ciencia de Materiales de Sevilla, Spain

Reply 2: Thanks for your useful suggestions. To address this point, we have added the previous works on helium bubbles and platelets in the **Introduction** section (see **Page 2**).

Following your suggestion, we have also provided a new **Discussion** section “Formation of helium platelets in diamond” (see **Pages 10, 11**) in the revised manuscript. In this section, we compared the blue shift of ~3.8 eV measured from the helium platelet (see **Fig. 1f**) with theoretical predictions of solid helium at ultra-high pressure. Based on self-consistent electronic structure calculations of solid fcc helium by Jäger et al. (ref. 53), we obtained the relation between energy shift ΔE and the inter-atomic distance R to consider the two-dimensional system:

$$\Delta E \text{ (eV)} = -0.8 + 31.1R^{-3}(\text{\AA}) \quad (\text{R1})$$

The theoretical blue shift of solid helium with respect to different inter-atomic distances is shown in **Fig. R1**. With $R = 1.8 \text{ \AA}$ given by iDPC STEM image and DFT calculations, equation (R1) yields the theoretical blue shift $\Delta E = 4.5 \text{ eV}$. The energy shift ΔE measured by EELS is reasonably smaller than the theoretical value because the $2p$ orbital of a fourfold coordinated atom in the two-dimensional structure will experience a weaker compressive effect than a helium atom having a larger number

(12) of closest neighbors in solid fcc helium.

Fig. R1 | Comparison of experimental blue shift of two-dimensional solid helium and theoretical blue shifts of solid fcc helium.

Response to Reviewer #2:

Comment 1: *“This manuscript contains some interesting novel experimental observations. In particular, the evidence for formation of solid platelets of He in implanted diamond and the accompanying reduction in the diamond bandgap by >2 eV are important observations. However, the scientific interpretation and discussion of the results should be strengthened. Several specific comments are listed below.”*

Reply 1: Thanks for your positive comment on our work.

Comment 2: *“Intro, p. 3. Why did the authors select a relatively extreme implantation temperature of 1573K? My understanding is that amorphization and graphitization can be suppressed in ion irradiated diamond at implantation temperatures of ~ 1370 K or lower. Some discussion of the temperature-dependent stability of diamond vs graphite should be provided here (and on p. 4).”*

Reply 2: Thanks for your useful comment. It is challenging to perform ion implantation into diamond at low temperature. We agree with the reviewer that amorphization and graphitization can be suppressed in the ion-irradiated diamond at implantation temperatures of ~ 1370 K or lower. We have conducted more than ten batches of experiments with different implantation temperatures ranging from room temperature to 1573 K. TEM observations demonstrated that helium ion implantation had little influence on the diamond structure at temperatures higher than 1173 K (see **Fig. R2** or **Supplementary Fig. 10**). However, high-temperature transformation of diamond to graphite will occur at temperatures higher than ~ 1873 K (ref. 47). As a result, in this work, we selected an implantation temperature of 1573 K to simultaneously suppress irradiation and thermal-induced graphitization in diamond. Moreover, such a high temperature can result in the formation of helium platelets with a larger diameter, which is conducive to high-quality S/TEM observations. Following your suggestion, we have added a new **Discussion** section “Temperature-dependent stability of irradiated diamond” (see **Pages 8-10**) in the revised manuscript.

Fig. R2 (Supplementary Fig. 10) | TEM characterization of diamond implanted at 1173 K. a, HRTEM image of the helium implantation peak region. **b,** Fast Fourier transform image taken from the region, revealing that amorphization and graphitization were suppressed. The zone axis (ZA) is [011].

Comment 3: “Larger font sizes should be used for axes labels and text in Fig. 1, 2, 3 and Supplementary Fig. 1b. (also typo in Suppl. Fig. 1b y axis label should be fixed; vacancies/cm⁻³ should be vacancies/cm³).”

Reply 3: Thanks for your careful review and useful comment. Following your suggestion, we have enlarged the font sizes in **Figs. 1, 2, 3** and **Supplementary Fig. 1b**. We have also corrected the y axis label in **Supplementary Fig. 1b**.

Comment 4: “Bottom p. 3: Why was a <001> oriented diamond crystal considered preferable to other orientations (<111>, etc.)? On p 4, it is noted that 001 platelets also formed in polycrystalline diamond for a <311> oriented grain. (relatively close to 001, vs. 110 or 111 orientation.

Based on prior studies on He-implanted ceramics such as silicon, SiC and Al₂O₃, platelets typically form on the low index plane closest to the surface orientation, and pronounced orientation effects for He-implanted diamond have been reported by other research groups (see M. Chen, et al., ref. 23; S.J. Zinkle, Nucl. Instrum. Meth. B 286 (2012) 4-19; and many other journal articles). Have the authors studied the effect of orientation (with respect to the irradiated surface) on formation of 2D platelets in diamond? Figs. 1b, 1c seems to show that platelets are located on 001 plane aligned with surface, whereas Fig. 1e shows some platelets on two perpendicular {001} habit planes; what was the relative fraction of platelets on the three {001} habit planes? (strongly anisotropic vs. nearly isotropic distribution?). Streaks in the SAED in Fig. 1d seem to be visible in only one {001} habit plane, suggesting that there was a strong anisotropy in the preferred habit plane. What was average platelet diameter? ”

Reply 4: Thanks for your insightful comment. The underlying reason for the {001} habit plane could be the contribution of elastic free energy of diamond to the total formation energy of helium platelets. Diamond is the most incompressible substance in nature, and its <100> direction is elastically softest (ref. 36). The formation of {001} helium platelets can result in the smallest elastically stored energy, and thus is thermodynamically favored in diamond. The same {001} habit plane has been also reported in helium-implanted silicon (refs. 48, 49), which has the same crystal structure and the similar elastic anisotropy with diamond.

We agree with the reviewer that in SiC, Si₃N₄, MgO, Al₂O₃ and MgAl₂O₄, platelets typically form on the low-index plane (*Nucl. Instrum. Meth. B* 286 (2012) 4-19). However, the formation of helium platelets in silicon occurs only in the {001} habit planes regardless of the orientation of the substrates, non-tilted and tilted (001)-, (110)-, and (111)-substrates (ref. 49). We didn't study the effect of orientation (with respect to the irradiated surface) on formation of helium platelets because we didn't receive the <011>-and <111>-oriented diamond. More related work will be conducted in the future and published in a separate paper.

The difference in habit planes between **Figs. 1b, c** and **Fig. 1e** is attributed to the different zone axes used in TEM imaging. Only (100) helium platelets can be observed under the zone axis of [011], while the zone axis of [001] allows us to observe platelets

with two perpendicular {100} habit planes. According to the statistical results from iDPC STEM imaging (see **Fig. R3a** or **Supplementary Fig. 5b**), the relative fraction of platelets on the (100) and (010) habit planes is about 2:1. It suggests that helium platelets in diamond have a strongly anisotropic distribution. It should be noted that we cannot directly observe all three {001} habit planes by our present experimental set-up. In future work, atomic electron tomography (AET) could be used to determine the three-dimensional structure of helium platelets and the relative fraction of platelets on the three {001} habit planes.

We counted the diameter of more than 200 helium platelets. The diameter distribution image is shown in **Fig. R3b** or **Supplementary Fig. 5a**, and the average platelet diameter is determined to be 4.6 nm.

Based on your suggestion, in the newly revised manuscript, we have provided the average platelet diameter and anisotropic distribution of platelets in the **Results** section and added a new **Discussion** section “Formation of helium platelets in diamond” to discuss the habit plane of platelets.

Fig. R3 (Supplementary Fig. 5) | Habit plane and diameter distribution of helium platelets in diamond. a, iDPC STEM image under the [001] zone axis, where (100) and (010) helium platelets are marked by yellow and blue lines, respectively. **b,** The diameter distribution of helium platelets in diamond.

Comment 5: “p. 5: what was typical spacing between 001 platelets? Is there any physical significance to this spacing? What controls the partitioning of the He into platelets? Is there a proposed reason why the He platelets reported by J. Chen et al. (1999) in SiC had a platelet width of ~2-3 atomic spacings, whereas your He-implanted diamond consisted of only one atomic layer?”

Reply 5: Thanks for your insightful comment. We measured the spacings between helium platelets by iDPC STEM imaging. The typical spacing between 001 platelets is 3.0 ± 1.3 nm (see **Fig. R4c** or **Supplementary Fig. 7c**), and the minimum platelet spacing is about 1 nm. It is expected that the spacing between 001 platelets is related

to helium pressure and can be controlled by implantation parameters (e.g., temperature and dose). Accordingly, to determine the relation between platelet spacing and helium pressure, we built up the diamond lattice with the inserted two-dimensional solid helium over a platelet spacing range from 1.07 to 2.14 nm (see **Fig. R4a** or **Supplementary Fig. 7a**) at the limits of computing. After fully relaxation of atomic positions via density functional theory (DFT) calculations, we obtained the supercell stress in different directions (see **Fig. R4b** or **Supplementary Fig. 7b**). It demonstrates that the interlayered pressure, i.e., P_{zz} , increases with decreasing platelet spacing and reaches its maximum of 166 GPa at 1.07 nm. The spacing is related to the inner pressure or compressive stress on the platelets from diamond lattice.

Fig. R4 (Supplementary Fig. 7) | Helium pressure and spacings between helium platelets. **a**, Atomic structural models of {100} two-dimensional solid helium with different platelet spacings. **b**, Pressure inside four models in (a) predicted by DFT calculations. **c**, Spacings between helium platelets measured by iDPC STEM imaging.

The partitioning of helium atoms into platelets is controlled by helium diffusion, and no matrix atom diffusion is required in the formation of helium platelets. Since diamond is the most incompressible substance in nature, it is energetically more

favorable for helium interstitial atoms to form helium platelets with only one atomic layer. The shear modulus of diamond (i.e., 553 GPa) is much higher than that of α -SiC (i.e., 192 GPa) reported by J. Chen et al. (ref. 23).

According to your insightful suggestion, we have added the relation between platelet spacing and helium pressure in the **Results** section and **Supplementary Fig. 7**, and added partitioning mechanism in the **Discussion** section “Formation of helium platelets in diamond”. Related computational details and discussions have also been revised.

Comment 6: *“top of p. 6. How was 56 GPa lattice pressure estimated? (this is not clear from the text in the methods section and Suppl. Fig. 7)”*

Reply 6: Thanks for your useful suggestion. To address the reviewers’ and editorial concerns, we have performed DFT calculations to determine the helium pressure inside platelets and determined the relation between platelet spacing and helium pressure (see details in **Reply 5**).

We have provided the helium pressure predicted by DFT calculations in the **Results** section and replaced the elastic continuum approach by DFT calculations in the **Methods** section accordingly.

Comment 7: *“top of p. 7. Are results consistent with previous reports of solid Ne, Ar, Kr platelets in implanted ceramics? (monolayer vs. multi-atomic thickness platelets). E.g., E.L. Fleischer, M.G. Norton. Noble gas inclusions in materials, Heterogeneous Chemistry Reviews 3 (1996) 171-201.”*

Reply 7: Thanks for your insightful suggestion. We have carefully read the review (*Heterogen. Chem. Rev.* 3 (1996) 171-201) and its references about solid inclusions in ceramics. We regret that the reported solid Ne, Ar, Kr and Xe inclusions are solid bubbles instead of solid platelets. These solid bubbles have an epitaxial relationship with the ceramic matrix. However, it is not rigorous to compare helium platelets with solid bubbles. Instead, we found the reports of solid neon (ref. 50) and xenon (ref. 51) platelets in metals, which are parallel to close-packed planes of the matrix. Our solid helium platelet results are inconsistent with solid Ne and Xe platelets in metals due to the elastic anisotropy of diamond (see more details in **Reply 4**).

We have added the comparison between helium platelets and solid neon and xenon platelets in the **Discussion** section “Formation of helium platelets in diamond”.

Comment 8: *“p. 7 Does the formation of He platelets produce compressive vs. tensile strains? Since the free surface can expand, large compressive lattice strain would not be expected in the matrix, whereas tensile strains would be expected to exist at the periphery of the platelets.*

Suppl. Fig. 7d seems to show high compressive strains within the platelets, and high tensile strains in the diamond lattice.”

Reply 8: Thanks for your insightful comment. We agree with the reviewer tensile strains exist at the edge of the platelets, which is similar with the strain field around a dislocation core. However, the top and bottom surface of helium platelets should produce large compressive strains to the surrounding diamond lattice, as predicted by the DFT calculations (see more details in **Reply 5**). The compressive strains in the diamond lattice are also confirmed by geometric phase analysis on the helium region. As shown in **Supplementary Fig. 9d**, most of the region is blue, indicating the compressive strains produced by helium platelets to the surrounding diamond lattice.

Comment 9: *“p. 8, last sentence: rewording needed. “achieving well-target and extraordinary properties””*

Reply 9: Thanks for your careful review of our work. According to your kind suggestion, we have reworded “achieving well-target and extraordinary properties” to “achieving tunable properties”.

Response to Reviewer #3:

Comment 1: *"In this work, two-dimensional solid helium is stabilized at room temperature within a diamond lattice matrix by means of ion-implantation techniques. The structure of the helium-based layers are characterized by experimental spectroscopy and computational DFT techniques.*

Despite that the technical quality of the present work may acceptable, I find that the present work does not advance any significant knowledge in the fundamental or applied sciences, as I explain in more detail next. In addition to such a lack of general interest and scientific relevance, I find that the authors fail in their attempt to assess the current state-of-the-art in the field of low-dimensional solid helium (or quantum solids). Therefore, I do not recommend publication of the present work in Nature Communications."

Reply 1: We appreciate your careful reading of our manuscript, and we respond to your concerns below.

Comment 2: *"As a matter of fact, two-dimensional solid helium has already been realized on top of common substrates like graphite [see, for instance, Journal of the Physical Society of Japan Vol. 77, No. 11, November, 2008, 111013 "Nuclear Magnetism in Two-Dimensional Solid Helium Three on Graphite" and Nature Physics volume 13, pages 455–459 (2017) "Intertwined superfluid and density wave order in two-dimensional 4He "] and carbon nanotubes [see, for instance, EPL, 88 (2009) 56005 "Dynamics of one-dimensional and two-dimensional helium adsorbed on carbon nanotubes"]. Therefore, there is not any novelty in stabilizing low-dimensional solid helium."*

Reply 2: Thanks for your useful suggestion. We agree that stabilizing two-dimensional solid helium has been achieved since the 1970s. In the initial manuscript, we have already provided some important literatures (refs. 12-16) in the **Introduction** section and compared our work with these studies (see **Fig. 4**). As pointed out by the reviewer, two-dimensional solid helium was only stabilized on the surface of common substrates (e.g., graphite and carbon nanotube) via physisorption at low temperatures. In the present work, one of our contributions is the introduction of the pressure dimension (i.e., a fundamental thermodynamic variable), substantially extending the P - T domain of two-dimensional solid helium. Our innovative work will allow the fundamental physics of two-dimensional solid helium to be examined at high pressure, which otherwise cannot be observed in previous systems. Moreover, compared to graphite and carbon nanotubes, diamond is more promising for semiconductor device applications (*J. Semicond.* 43 (2022) 021801). Engineering diamond by two-dimensional solid helium opens up a new avenue for optimizing diamond semiconductor.

Comment 3: *"The interest of studying two-dimensional solid helium at low temperatures ($T \sim 1\text{K}$) is purely fundamental: helium is the archetypal quantum crystal and lowering its dimensionality may lead to new quantum atomic physics and exotic*

states of matter (e.g., helium atoms are boson particles and can form Bose-Einstein condensates). However, there is not any fundamental interest in studying low-dimensional solid helium at room temperature and high pressures since under those thermodynamic conditions the quantum nature of solid helium is totally absent and the system behaves classically (i.e., no atomic delocalization, no Bose-Einstein condensation, no exotic states of matter can emerge -see, for instance, *Rev. Mod. Phys.* 89, 035003 (2007) "Simulation and understanding of atomic and molecular quantum crystals"). On the other hand, I fail to appreciate any practical interest and relevance out of synthesizing two-dimensional solid helium in a diamond matrix; on this regard, the authors mention some possible "intrinsic strain doping" but that is something that could be equally achieved by using other atomic light species (e.g., Li) and same ion implantation techniques (...probably even more easily than with helium!)."

Reply 3: Thanks for your useful comment. We apologize that we did not present it very clear. Here we would like to clarify these two points: (1) **Fundamental interest.** We agree with the reviewer that the quantum nature of solid helium only exhibits at sufficiently low temperature. In this work, we focused on realizing two-dimensional solid helium stabilized by high pressure. However, our sample can be cooled to low temperatures ($T \sim 1\text{K}$) for exploring quantum atomic physics and exotic states of matter. Moreover, solid helium under high-temperature-pressure conditions is of great fundamental interest to astrophysicists (*Phys. Rev. Lett.* 101 (2008) 106407; ref. 1) and geologists (*Nat. Geosci.* 6 (2013) 982-986; *Nat. Commun.* 12 (2021) 1-11.); (2) **Practical interest.** By implanting helium ions to form two-dimensional solid helium, we introduced large elastic strains into diamond that could substantially change its electronic properties for potential semiconductor applications. By contrast, it is difficult to realize our strain doping strategy by using other atomic light species. Implanted Li atoms usually occupy interstitial sites in the diamond lattice and then form clusters (*Diam. Relat. Mater.* 4 (1995) 862-872), which can only generate non-uniform and localized elastic strains. Although other interstitial light species (e.g., Li) can give rise to n-type doping of diamond, it is difficult to incorporate Li atoms into the diamond lattice and the donor levels are shallow (*Diam. Relat. Mater.* 10 (2001) 1749-1755; *Diam. Relat. Mater.* 16 (2007) 840-844).

Comment 4: "In addition to these lacks of novelty and scientific and practical interests, I find that some of the claims made by the authors in their manuscript are not sufficiently well justified. For instance, in the abstract it appears written that solid helium monolayers were observed at "ultra-high pressures up to 56 GPa". This information gives the impression to the readers that the authors have performed some high-pressure diamond anvil cell experiments (DAC) under well-defined hydrostatic pressure conditions. However, one rapidly realizes in reading the subsequent sections that such a claim is quite misleading since it is based on a very crude elastic approximation (explained in Sec. "Helium pressure imposed by the diamond lattice") that completely disregards any atomistic detail. [...Actually, how a ultrahigh pressure of 56 GPa can be maintained in a stable manner in the interior of a matrix that as a whole is in equilibrium at ambient pressure?? To most scientists versed on thermodynamics this statement does not make any physical sense...]. Moreover, the

exact crystal structure of the stabilized two-dimensional solid helium is not satisfactorily characterized (i.e., the corresponding cif file appears to be missing).

For all these reasons, unfortunately, I cannot be more positive in my evaluation.”

Reply 4: Thanks for your constructive comment. To avoid misunderstanding, we have reworded “under ultra-high pressures up to 56 GPa” to “compressed by the robust diamond lattice” in the **Abstract** section. To further address the reviewers’ and editorial concerns, we have considered the relation between platelet spacing and helium pressure and performed DFT calculations to determine the helium pressure inside platelets (see details in **Supplementary Fig. 7**).

From a thermodynamic point of view, the free energy of a helium platelet (see schematic atomic structure in **Fig. R4**) includes the surface/interface free energy, the elastic stored energy associated with the elastic crack and helium atoms contained in a platelet. The elastic stored energy results from the helium atom induced displacement (see red arrows in **Fig. R4**). In other words, helium atoms in a crack (i.e., a platelet) are under high pressure imposed by diamond lattice with high elastic strength.

Fig. R4 | Atomic structure of a helium platelet in diamond lattice. Red arrows highlight the lattice displacement of diamond lattice induced by a helium platelet.

According to your suggestion, we have provided the exact crystal structure (i.e., cif file) as the supplementary material in the newly revised manuscript. The corresponding cif file of the stabilized two-dimensional solid helium (**Fig. 2c**) is also listed in the following.

```
#=====
# CRYSTAL DATA
#-----
data_VESTA_phase_1

_chemical_name_common      'This file is generated by VASPKIT
code'
_cell_length_a              2.519350
_cell_length_b              2.519350
_cell_length_c              14.251600
_cell_angle_alpha           90.000000
_cell_angle_beta            90.000000
_cell_angle_gamma           90.000000
_cell_volume                 90.456684
_space_group_name_H-M_alt   'P 1'
_space_group_IT_number      1
```

```
loop_  
_space_group_symop_operation_xyz  
  'x, y, z'
```

```
loop_  
_atom_site_label  
_atom_site_occupancy  
_atom_site_fract_x  
_atom_site_fract_y  
_atom_site_fract_z  
_atom_site_adp_type  
_atom_site_U_iso_or_equiv  
_atom_site_type_symbol  
He1      1.0      0.000000      0.500000      0.492960      Uiso  ? He  
He2      1.0      0.500000      0.000000      0.507040      Uiso  ? He
```

REVIEWERS' COMMENTS

Reviewer #1 (Remarks to the Author):

Authors have carefully revised the manuscript considering reviewer's comments and questions. I recommend the paper for publication

Reviewer #2 (Remarks to the Author):

The authors have generally provided acceptable responses to the referee queries and made several useful modifications to the manuscript (a few of the other comments made in the referee response document would have been useful to mention in the revised manuscript). I do not have further mandatory suggested changes.

Reviewer #3 (Remarks to the Author):

The authors have addressed quite satisfactorily most of my previous concerns and clarified few important points in their article revision. I am still not fully convinced about the practical and fundamental interests of the present work; however, based on the reports by all the Reviewers and the authors responses to their queries, I may simply be missing some relevant aspects on my evaluation (which always turns out to be a bit subjective as it depends on the specific background of the Reviewer). Therefore, trying to be as reasonable as possible, I do recommend publication of the present work.

I only have one final recommendation to the authors. In order to emphasize the possible fundamental interests in stabilizing two-dimensional solid helium without the need of employing high-pressure techniques (e.g., for exploring quantum atomic physics and exotic states of matter at very low temperatures), the authors may consider including the following references in their article, which are seminal works in the field of pressurized and low-dimensional quantum crystals:

- Reviews of Modern Physics 89, 035003 (2017)
- Nature Physics 13, 455 (2017)
- Reviews of Modern Physics 84, 1607 (2012)

Response to the Reviewers

Response to Reviewer #1:

Comment: *“Authors have carefully revised the manuscript considering reviewer’s comments and questions.*

I recommend the paper for publication.”

Reply: We feel great thanks for your professional review work on our manuscript.

Response to Reviewer #2:

Comment: *“The authors have generally provided acceptable responses to the referee queries and made several useful modifications to the manuscript (a few of the other comments made in the referee response document would have been useful to mention in the revised manuscript). I do not have further mandatory suggested changes.”*

Reply: We are greatly appreciative of the effort and time from the reviewer to provide helpful insights.

Response to Reviewer #3:

Comment: *“The authors have addressed quite satisfactorily most of my previous concerns and clarified few important points in their article revision. I am still not fully convinced about the practical and fundamental interests of the present work; however, based on the reports by all the Reviewers and the authors responses to their queries, I may simply be missing some relevant aspects on my evaluation (which always turns out to be a bit subjective as it depends on the specific background of the Reviewer). Therefore, trying to be as reasonable as possible, I do recommend publication of the present work.*

I only have one final recommendation to the authors. In order to emphasize the possible fundamental interests in stabilizing two-dimensional solid helium without the need of employing high-pressure techniques (e.g., for exploring quantum atomic physics and exotic states of matter at very low temperatures), the authors may consider including the following references in their article, which are seminal works in the field of pressurized and low-dimensional quantum crystals:

- Reviews of Modern Physics 89, 035003 (2017)

- Nature Physics 13, 455 (2017)

- Reviews of Modern Physics 84, 1607 (2012)”

Reply: Thanks for your useful suggestion. We have added the additional references in the **Introduction** section of the revised manuscript (Page 2).